# Role of Capillaroscopy in Early Diagnosis of Ionizing Radiation Damage in Healthcare Professionals

**DOI:** 10.3390/medicina59071356

**Published:** 2023-07-24

**Authors:** Luca Di Bartolomeo, Federica Li Pomi, Francesco Borgia, Federico Vaccaro, Fabrizio Guarneri, Mario Vaccaro

**Affiliations:** 1Section of Dermatology, Department of Clinical and Experimental Medicine, University of Messina, 98125 Messina, Italy; 2Department of Dermatology, University of Modena and Reggio Emilia, 41124 Modena, Italy

**Keywords:** capillaroscopy, ionizing radiation, healthcare professionals, chronic radiation exposure

## Abstract

*Background and Objectives*: Chronic ionizing radiation has biological effects on exposed healthcare workers, particularly on the skin. Capillaroscopy of the nail bed represents an easy, low cost, and non-invasive test to obtain information on the effects of chronic radiation exposure in healthcare workers. The aim of this study was to evaluate which capillaroscopic parameters are most associated with biological damage by chronic radiation exposure. *Materials and Methods*: We conducted a case-control study, in which cases were represented by healthcare workers exposed to ionizing radiations and controls by healthy subjects. We recorded anamnestic and personal data, including age and gender, before capillaroscopic examination of proximal nail folds of the fingers of both hands. Ten morphological qualitative/quantitative parameters were taken into consideration, assigning each of them a score on a scale from 0 to 3 (0 = no changes, 1 = <33% abnormal capillaries, 2 = 33–66% of abnormal capillaries, 3 = >66% of abnormal capillaries, for single magnification field at 200×). The parameters evaluated were: changes in the length, distribution and density of capillary loops, reduced visibility, decreased flow, visibility of the sub-papillary plexus, and presence of morphological atypia, such as ectasia, tortuosity, hemorrhage, and signs of neoangiogenesis. *Results*: We enrolled 20 cases and 20 controls. The two groups did not differ significantly for gender and age. Cases differed from controls in a statistically significant way for the following parameters: decreased capillary length (number of shortened capillaries) (*p* < 0.05), increased visibility of the subpapillary venous plexus (*p* < 0.05), tortuosity (*p* < 0.01), neoangiogenesis (*p* < 0.01), and ectasias (*p* < 0.001). *Conclusions*: We found that some capillaroscopic parameters, such as variability in length of capillaries, visibility of subpapillary venous plexus, presence of ectasias, tortuosity, and neoangiogenesis signs, are particularly associated with exposure to ionizing radiation in healthcare professionals. Alterations of these parameters may represent capillaroscopic clues of biological damage by chronic radiation exposure in healthcare professionals. Based on these observations, capillaroscopy may provide clinical data useful to the prevention and follow-up of radiation-exposed healthcare professionals.

## 1. Introduction

Exposure to ionizing radiation implies a certain risk of biological damage in the short and long term. Biological effects of ionizing radiations are relevant not only in directly exposed patients but also in healthcare workers who are indirectly exposed to them. Their increasing use in diagnostic and therapeutic procedures has raised the need of quantifying biological damage in healthcare professionals and ensuring them an adequate level of radioprotection. Several European studies have suggested that current dosimetry procedures may lead to an underestimate of dose levels, particularly at the level of hands, which are the most exposed areas [1]. Considering that vascular microcirculation is one of the first targets of ionizing radiation damage, capillaroscopy of the proximal nail fold has been used to assess the effects of chronic ionizing radiation on healthcare professionals. Capillaroscopy is a simple, non-invasive, and cheap examination that allows to obtain clinical data regarding systemic diseases with observations made on the capillaries of the proximal nail fold. One of the most common uses of capillaroscopy is to distinguish between primary and secondary Raynaud’s phenomenon [2]. Nevertheless, capillaroscopy has been demonstrated useful in monitoring several diseases, such as obstructive sleep apnea or, recently, COVID-19 [3,4]. The purpose of this study is to add a piece in the definition of a capillaroscopic pattern of ionizing radiation damage, by identifying which capillascopic parameters are most affected in healthcare professionals chronically exposed to ionizing radiation. The definition of a capillaroscopic score of ionizing radiation damage is challenging and a previous study attempted to reach it [5]. However, our study took into consideration more parameters than the previous one and found more significant capillaroscopic changes in radiation-exposed workers.

## 2. Materials and Methods

### 2.1. Patients and Study Design

We carried out a case-control study, in which cases were represented by subjects exposed to ionizing radiations (radiologists, nuclear medicine physicians, radiology technicians) and controls by healthy subjects. The enrollment of cases took place in the Unit of Occupational Medicine of the University Hospital “G. Martino” of Messina, while controls were enrolled in the Unit of Dermatology and Venereology of the same hospital. The inclusion criteria were:-for cases: adult men and women, exposed to ionizing radiation for professional reasons, in apparent good state of health;-for controls: adult men and women, with a negative personal history of exposure to ionizing radiation and in apparent good state of health.


Exclusion criteria for cases and controls included:-personal and/or family history of autoimmune diseases and connective tissue diseases, recent traumas, hand dermatitis, or other diseases with possible microvascular impact (neoplasms, hypertension or diabetes);-performing nail cosmetic procedures (manicure) or applying nail polish or gel in the 30 days before capillaroscopic investigation;-presence of radiodermatitis, at the time of the visit or within the previous 30 days.

Subjects meeting the above criteria were informed about the methods and aims of the study and invited to participate. All subjects participating in the study signed informed consent for execution of capillaroscopy and data processing. Anamnestic and personal data, including age and gender, including presence/absence of Raynaud’s phenomenon, of all subjects were collected before capillaroscopic examination.

### 2.2. Capillaroscopic Examination

Enrolled patients underwent a capillaroscopic examination of proximal nail folds of the fingers of both hands. For reasons related to feasibility of the diagnostic investigation, the capillaroscopic exam was conducted on all fingers except thumbs.

The examination was conducted with a “Videocap” video-capillaroscope (DS-Medica, Milan, Italy) with 100× and 200× optical probes, connected to a digital data storage system. One or more images were recorded for each finger examined and archived in a database.

### 2.3. Capillaroscopic Parameters

Given the absence of a specific capillaroscopic pattern caused by ionizing radiation, 10 morphological qualitative/quantitative parameters were taken into consideration, assigning each of them a score on a scale from 0 to 3 (0 = no changes, 1 = <33% abnormal capillaries, 2 = 33–66% abnormal capillaries, 3 = >66% abnormal capillaries, for single magnification field at 200×). The parameters evaluated were: changes in the length, distribution and density of capillary loops, reduced visibility, decreased flow, visibility of the sub-papillary plexus, and presence of morphological atypia, such as ectasia, tortuosity, hemorrhage, and signs of neoangiogenesis. In the healthy subject, the length of the capillaries is homogeneous within the same field of observation, although it can vary from finger to finger. In general, the length of the capillary loops is between 200 and 300 μm [6,7]. The diameter of the capillaries varies between the arterial (afferent) and venous (efferent) tracts. In fact, the former is generally narrower (8–10 μm), while the latter is wider (10–14 μm) [6]. In general, capillary dilation is considered when increased capillary diameter is >20 μm [8]. Regarding the capillary density, in one millimeter of nail fold, 7 to 17 capillaries are contained [6,7]. Regarding tortuosities, in a normal capillaroscopic pattern, less than 2 tortuosities per mm are present [7]. 

### 2.4. Statistical Tests

All data obtained were used for descriptive analyses. The variables considered for statistical analysis were: gender, age, and capillaroscopic parameters. Values relating to gender were summarized by absolute number and frequency, while those relating to age were by mean, standard deviation, minimum, and maximum. Capillaroscopic parameter values were expressed as ordinal variables on a scale from 0 to 3, and absolute number of occurrences and frequency of each value was recorded. Differences between cases and controls were analyzed by means of the chi-square test (χ^2^) for gender, two-tailed Student’s *t*-test for age, and Fisher’s exact test for capillaroscopic parameters. The pre-established significance level was α = 0.05, thus *p*-values less than 0.05 were considered statistically significant for 2-sided tests.

## 3. Results

### 3.1. Clinical and Anamnestic Results

The subjects recruited in the 2 groups were 20 cases and 20 controls, respectively. The male/female ratio was 9/11 among cases and 11/9 in the control group. Therefore, there were no significant differences between cases and controls as regards gender (*p* = 0.53). The mean age in the case group was 43.55 ± 13.43 years (minimum age 27 years, maximum age 63 years), while, in the control group, it was 38.7 ± 11.6 years (minimum age 21 years, maximum age 60 years). Therefore, the 2 groups did not differ significantly for age (*p* = 0.23). From an anamnestic point of view, Raynaud’s phenomenon was absent in all subjects; 5 patients in the case group and 3 patients in the control group had other comorbidities, including dyslipidemia, polycystic ovary syndrome, osteoarthritis, osteoporosis, hearing impairment, and cataract. Table 1 summarizes the clinical and anamnestic features of the 2 groups.

### 3.2. Capillaroscopic Parameters in the Two Groups

The group of cases showed more relevant microcirculation alterations than the control group. In particular, cases differed from controls in a statistically significant way for the following parameters: decreased capillary length (number of shortened capillaries) (*p* < 0.05), increased visibility of the subpapillary venous plexus (*p* < 0.05), tortuosity (*p* < 0.01), neoangiogenesis (*p* < 0.01), and ectasias (*p* < 0.001). Table 2 summarizes the results of Fisher’s exact test for each capillaroscopic parameter.

Figure 1 compares the capillaroscopic findings found in radiation-exposed subjects and healthy controls.

## 4. Discussion

Healthcare workers employed in the field of radiology and radiotherapy are daily subjected to low doses of ionizing radiation, whose impact on long-term health is being studied. In the context of stochastic damages from radiation and, in particular, for doses lower than 100 mGy, it is not possible to exclude the onset of cancer, and furthermore, below this threshold, the incidence of hereditary diseases or tumors seems proportional to the equivalent dose increment [9]. Epidemiological data directly suggest increased cancer risk in the 10 mSv to 100 mSv range. There is an excess risk of developing cancer of 5% per Sv and this can be linearly extrapolated for lower doses [10]. In the early 2000s, Sutherland and colleagues demonstrated that even low doses (from 0.1 to 1 Gy) of radiations with a high-linear energy transfer (LET), in particular deriving from Fe^26+^ ions, may cause DNA alterations in human cells with an index survival greater than 90% [11]. Although stable chromosomal aberrations cannot be found at doses below 100 mGy [12], the likelihood of these types of mutations increases with the amount of radiation absorbed over time during occupational exposure [13]. Little and colleagues studied the presence of chromosomal aberrations in 282 radiology technicians using fluorescent in situ hybridization (FISH). The authors observed an increasing relationship between dose and number of chromosomal aberrations for estimated cumulative exposures less than 100 mSv. The presence of mutations or radio-induced chromosomal aberrations does not necessarily imply clinical effects, and exposure to low doses does not cause clinical symptoms, but may cause chronic effects over time. The first clinical manifestations of chronic radiation damage are usually localized to the extensor surface of the hands, particularly of the second and third phalanges, also involving the nail bed [9]. In fact, hands of healthcare workers are the areas of body most susceptible to radiation risk because are the least covered but the closest to the sources of radiation. Furthermore, personal dosimetry systems have limits to correctly estimating the radiation doses in the hands, being these systems often are positioned at the chest. For dosimeters measuring doses at the extremities, i.e., bracelet or ring ones, there are no unequivocal indications on the position where they should be worn.

The ORAMED (Optimization of RAdiation protection for MEDIcal staff) project, promoted by the European Union, was conducted by numerous study groups from February 2008 to January 2011 [14] to study radiation exposure in healthcare professionals performing interventional radiology and/or cardiology procedures or nuclear medicine physicians. Several study groups belonging to the ORAMED project studied dosimetry in interventional radiologists and cardiologists [15,16,17,18,19]. In this regard, Domienik and colleagues found that the “kerma area product (KAP)”, also called “dose area product (DAP)”, i.e., the absorbed dose multiplied by the irradiated surface area, could highly differ in healthcare workers depending on the procedures, ranging from 40 to 271,000 microGy/m^2^. In particular, the highest values were recorded during angiographic procedures while the lowest ones were referred to pacemaker and defibrillator implantation or cholangiopancreatography [15]. In subsequent work, Domienik and colleagues studied absorbed doses at the eye and extremities in workers employed in interventional cardiology departments. Dosimeters were placed near eyebrows and bilaterally at fingers, wrists, knees, and ankles. As regards fingers, the authors recorded values from 0.007 to 2.25 mSv, again confirming the extreme variability of measurements. Furthermore, the authors estimated the annual exposure doses, finding that they were lower than the recommended limits for fingers; nevertheless, the maximum values were considerably close to the recommended limits, exceeding half the limit dose of 500 mSv for 1 cm^2^ of skin (estimated maximum annual dose: 355 mSv). On the other hand, the maximum estimated annual doses referred to the eye’s lens reached 247 mSv, exceeding the limit dose of 150 mSv (valid from 1996 until 2013, when the European directive 2013/59/EURATOM lowered the annual dose limit for the eye’s lens to 20 mSv) [20,21]. The authors underlined that dose levels in healthcare workers may vary according not only to the type of radiation but also to the availability of protective devices and position with respect to the radiant source [16]. Furthermore, as confirmed by other studies [17,18], catheter access in angiographic procedures influences the degree of exposure.

In their work on dosimetry in interventional radiology departments in 6 different European countries, Nikodemová and co-authors studied the personal protective equipment used by healthcare workers, highlighting that most of them use both lead aprons and collars and only 25% use also protective glasses. Up to 1% of healthcare workers do not use any personal protective measures. Regarding environmental protection measures, the surprising fact is that almost a quarter of workers (23%) do not use radiation protection measures in the operating room [19]. One of the aims of the ORAMED project was the study of radiation exposure in nuclear medicine workers. In this regard, Sans Merce and colleagues carried out a study in 32 nuclear medicine departments in Europe on workers involved in the preparation and administration of the 2 most used radionuclides for diagnostic purposes, namely 99mTc and 18F, and of 90Y-based radiopharmaceuticals [1]. The authors studied exposures of workers’ hands, including wrists, recording skin doses ranging from 0.07 to 32.05 mSv per Giga-Becquerel (GBq). In the diagnostic setting, doses were higher for 18F than 99mTc, although diagnostic procedures involving 99mTc are usually more frequent than 18F. Compared with diagnostic procedures, the therapeutic procedures determined a higher exposure and doses were higher in the preparation phase than in radiopharmaceutical administration. The authors explained this difference considering three reasons: manipulation activities during preparation are more numerous than those needed for administration of the radiopharmaceutical; secondly, some preparation steps are performed with an unshielded source, while administration of the radiopharmaceutical is usually performed with a shielded syringe; and finally, the time required to prepare a radiopharmaceutical is longer than the time required to administer it. The authors found the greatest exposure at the forefinger and thumb, most often of the non-dominant hand, while the wrist position, which is usually used to monitor exposure of the extremities, was the least exposed area of the whole hand. In particular, fingertips were more exposed than the base of the index or ring finger, where the dosimetric measurement of the hand is usually carried out. Based on this observation, it is suggested that the measurements at the tip of the finger may be higher than at the base of the finger, thus exceeding the annual dose limit. Therefore, the most correct position for dosimeters should be the tip of the index finger of the non-dominant hand. Since this measurement is difficult in daily practice, one could be satisfied with the measurement made at the base of the index, considering however that it is 2.5 times smaller than the maximum cutaneous dose at the tip [1]. In the absence of a fingertip dosimetry, capillaroscopy of the nail bed represents a feasible test to obtain information on the effect of radiation exposure in healthcare workers. The advantages of capillaroscopy are related to the non-invasiveness, the relative ease of execution and repeatability, the immediate reading of the results, and the low cost. Moreover, modifications of the nail bed represent one of the first signs of radiation damage, often in the absence of other clinical signs [9]. The major limitation of capillaroscopy is the lack of a specific capillaroscopic pattern of radiation damage. The first studies on capillaroscopy in the field of radiation damage appeared from the second half of the 1980s [22]. In 1996, Tomei and colleagues performed a case-control capillaroscopic study of 146 radiation-exposed physicians including cardiologists, radiologists, and orthopedists to evaluate the damage on microcirculation due to low-dose exposure [9]. Notably, annual finger exposure in physicians did not exceed a dose of 60 rem (600 mSv). The authors analyzed capillary damage entity, blood flow, changes in capillary caliber and length, and the presence of coils, tortuosity, and hemorrhages. A total of 20.7% of exposed subjects presented a normal capillaroscopic picture, versus 91.5% of controls. Further, 79.3% of exposed workers had damage patterns ranging from severe (12.4%) to mild (26.9%), compared with less than 1% of moderate-severe patterns in the control group. The authors noted that the prevalence of capillaroscopic damage was closely related to years of work, with a statistically significant difference between subjects who had a history of occupational exposure greater than or less than 20 years. Unfortunately, the authors failed to find specific alterations in the exposed subjects, as capillaroscopic alterations of the control group did not differ from those found in the exposed subjects. The absence of a capillaroscopic radiation injury score remains one of the key points in this research field [9]. Recently (2016), Wild et al. carried out a study similar to that of Tomei’s group, involving 186 radiation-exposed healthcare workers and 35 physicians as the control group [5]. The French authors tried to validate the effectiveness of two quantitative scores, one concerning the aspects of blood extravasation and the other the morphological modifications, based on seven semi-quantitative indices. The score concerning blood extravasation was based on the evaluation of image visibility degree as an expression of edema severity, the number of visible capillary loops, and their length. Instead, the score concerning morphological modifications took into consideration the irregularity of distribution of capillaries, the presence of tortuosity, the severity and number of dystrophies, and neo-angiogenesis signs. A score from 0 to 2 was assigned to all indices depending on the severity of changes. Furthermore, subjects participating in the study answered a questionnaire regarding their occupational exposure, making it possible to also have an assessment of the weekly and cumulative exposure during their entire working life. The results of this study did not allow to make evaluations on the effect of exposures based on the blood extravasation score. On the other hand, the score based on morphological abnormalities revealed that the severity of changes was proportional to the duration of exposure and estimated cumulative exposure among interventional radiologists and surgeons. The same effect was not seen among radiation-exposed cardiologists, who did not show a morphological score different from that of unexposed subjects. This result was a surprise for the authors, since the frequency of weekly exposures of cardiologists, as well as time and dose of exposures, were higher than those of the other groups. The authors did not find unequivocal explanations for this phenomenon but supposed that it was due to the diversity of procedures. In fact, only 7% of the cardiologists examined carried out activities at close range to radiation, versus 55% of radiologists. The authors concluded by underlining that, despite possible biases related to performance and interpretation, capillaroscopy could represent a valid tool for secondary prevention and follow-up of radiation-exposed subjects, even though large longitudinal studies are needed to validate this approach [5].

Our study, conducted on 20 healthcare professionals exposed to ionizing radiation and 20 healthy controls representing 2 groups not dissimilar in age and gender, highlighted that the 2 groups differed significantly for some capillaroscopic variables. Subjects exposed to ionizing radiation, compared to healthy controls, showed more accentuated alterations in the length of capillaries, specifically shortening of loops, greater visibility of the subpapillary venous plexus, a greater frequency of ectasias, tortuosity, and signs of neoangiogenesis. Differences between the two groups were also found in other parameters including the presence of bleeding, but these differences did not reach the minimum level of statistical significance required. Instead, among the statistically different parameters between the two groups, tortuosity, signs of neoangiogenesis, and ectasias were distinguished by a higher level of statistical significance. This could lead to hypothesize that modifications of these parameters are associated with radiation damage, but this hypothesis should be confirmed by further studies. Certainly, dilatation of capillaries represents the first sign of damage of the microvessel wall, since it is an expression of alteration of the endothelial cell matrix. This could explain why ectasias have the highest statistical significance among all the parameters considered (*p* = 0.00001).

Although our study differs in part from those previously carried out on the topic [5,9], our results seem to partially confirm those of other study groups. Both the French study by Wild and colleagues and ours highlighted a higher frequency of capillary tortuosity and dystrophies in healthcare professionals exposed to ionizing radiation. However, the French study did not find statistically significant differences in length of capillaries or neoangiogenesis signs, nor it analyzed some parameters taken into consideration in our study, namely visibility of the subpapillary venous plexus. In our study, we recorded a similar reduction in visibility and density of capillaries in cases and controls. However, the relative similarity between the two groups could also be explained by external factors, including errors in performing the capillaroscopic examination: images with poor visibility or apparent reduction in density of capillaries even in healthy subjects could have been due to poor focus rather than intense edema. Other limitations of our study must also be considered, which may be due to the characteristics of patients or to the operators who conducted and evaluated capillaroscopic examinations. Numerous factors with an effect on nail microcirculation were not investigated in the clinical history collection of the study. These include the habit of smoking or nail biting. The latter habit, as well as the occurrence of unrecognized microtraumas, could have influenced the presence of microhemorrhages even in healthy subjects. Finally, operator-dependent evaluation errors, such as correct grading of the capillaroscopic parameters considered (for example, evaluation of length or tortuosity degree of capillaries), could also have influenced the data analysis. Therefore, the results of the study must be interpreted considering these limitations, although some data are so significant that they are highly unlikely to be explained by errors or random effects.

## 5. Conclusions

We conducted a case-control study aimed at evaluating the biological effects of chronic radiation exposure in healthcare professionals by capillaroscopic examination of proximal nail folds. We found that, compared with controls, healthcare workers chronically exposed to ionizing radiation presented decreased capillary length, increased visibility of the subpapillary venous plexus, and higher frequency of ectasias, tortuosity, and neoangiogenesis signs. These capillaroscopic alterations may be clues of radiation damage and could be part of a future definition of a specific capillaroscopic picture in the early stages of biological damage in this group of healthcare workers. The definition of a capillaroscopy pattern for this condition may help to improve prevention and follow-up in radiation-exposed healthcare professionals.

## Figures and Tables

**Figure 1 medicina-59-01356-f001:**
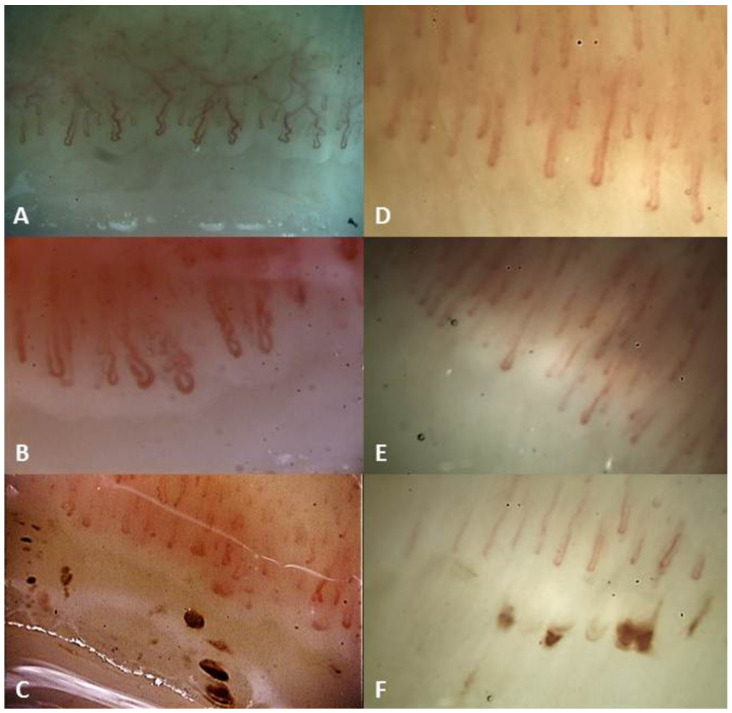
(**A**–**C**): Capillaroscopic images in healthcare professionals exposed to ionizing radiation; (**D**–**F**): capillaroscopic images in healthy controls. Capillaroscopy showed that tortuosities of capillaries were more prominent in subjects exposed to ionizing radiation than in healthy controls (**A**,**D**), as well as the subpapillary venous plexus was more visible in the former than in the latter. Subjects exposed to ionizing radiation presented more frequently loop ectasias (**B**,**E**), while there were no significant differences between the two groups as regards the presence of hemorrhages (**C**,**F**).

**Table 1 medicina-59-01356-t001:** Clinical and anamnestic features of the two groups, healthcare workers exposed to ionizing radiation and controls.

	Cases	Controls	*p*-Value
Gender	9 males, 11 females	11 males, 9 females	0.53
Age	43.55 ± 13.43 years	38.7 ± 11.6 years	0.23
Comorbidity	5	3	

**Table 2 medicina-59-01356-t002:** Results of Fisher’s exact test for each capillaroscopic parameter considered in the study. The first column presents the capillaroscopic parameters considered in both groups. The second and third columns present the data used to create the contingency tables, namely the number of subjects who obtained a specific score on a scale from 0 to 3 for each capillaroscopic parameter (0 = no changes, 1 = <33% abnormal capillaries, 2 = 33–66% abnormal capillaries, 3 = >66% abnormal capillaries, for single magnification field at 200×). The last column presents the *p* values, which are indicated with asterisks when statistically significant (* *p* < 0.05, ** *p* < 0.01, *** *p* < 0.001).

Capillaroscopic Parameters	N. of Cases with Scores from 0 to 3 (0; 1; 2; 3)	N. of Controls with Scores from 0 to 3 (0; 1; 2; 3)	*p*-Value
changes in capillary length	1; 5; 10; 4	8; 6; 5; 1	0.025 *
capillary distribution	4; 8; 6; 2	6; 5; 6; 3	0.72
capillary density	1; 6; 7; 6	3; 8; 5; 4	0.66
reduced visibility	1; 4; 9; 6	3; 3; 9; 5	0.87
decreased flow	2; 4; 13; 1	6; 6; 8; 0	0.18
visibility of the sub-papillary plexus	1; 7; 11; 1	7; 9; 3; 1	0.014 *
ectasias	1; 7; 10; 2	15; 4; 1; 0	0.00001 ***
tortuosity	1; 2; 9; 8	9; 3; 7; 1	0.0044 **
hemorrhages	2; 4; 10; 4	8; 6; 4; 2	0.073
signs of neoangiogenesis	2; 8; 9; 1	13; 5; 2; 0	0.00116 **

## Data Availability

The data presented in this study are available, on request, from the corresponding author.

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
