# Peer review of "Role of Capillaroscopy in Early Diagnosis of Ionizing Radiation Damage in Healthcare Professionals"

_medicina, 2023, doi:10.3390/medicina59071356_

Round 1

Reviewer 1 Report

The issue Presented in the manuscript is relevant to everyday clinical practice. The methodology of the study and ethical conduct of the Authors do not raise any objections. The collected data is analysed and presented in a clear and transparent way. Conclusions drawn from the results are logical. The presented diagnostic approach can be easily applied into routine clinical practice.

While the manuscript could be published in the current form it would be interesting to know what was the actual radiation exposure of the cases (e.g. the average annual absorbed dose expressed in mSv from dosimetry readings) and weather there is any correlation between the absorbed dose and the extent of capillary changes.

There is a small typo - in line 28 the word 'visibility' is unnecessarily repeated.

Author Response

Dear Reviewer,  

Thank you for your encouraging comments, we are pleased that you appreciated our manuscript. In this preliminary study, we evaluated only capillaroscopic alterations in subjects exposed to ionizing radiation regardless of the average dose and/or exposure time. These evaluations (the average annual radiation exposure of the cases and if there is any correlation between the absorbed dose and the extent of capillary changes) will be carried out in a next study based on larger number of partecipants which we will shortly submit.  We corrected the typo in line 28.  

Reviewer 2 Report

The topic is interesting, the paper quite well written. I have few comments:

1) Abstract. L20- 22. Ten morphological qualitative/quantitative parameters were taken into considera-tion, assigning each of them a score on a scale from 0 to 3 (0 = no changes, 1 = <33% abnormal ca-pillaries, 2 = 33–66% of abnormal capillaries, 3 = >66% of abnormal capillaries, for single magnifi- cation field at 200x). Could you please insert the ten parameters evaluated?Do you evaluate the number of capillaries?

2) Abstract. L28-30. Conclusions: We found that some capillaroscopic parameters, such as variability in length of capillaries, visibility of subpapillary venous plexus visibility, presence of ectasias, tortuosity and neoangiogenesis signs, are particularly associated with exposure to ionizing radiation in healthcare professionals. Alterations of these parameters may represent capillaroscopic clues of biological damage by chronic radiation exposure in healthcare professionals. Could you please improve the description of possible implications of these observations?

3) 1. Introduction L35-37. Exposure to ionizing radiation implies a certain risk of biological damage in the short and long term. Biological effects of ionizing radiations are relevant not only in directly exposed patients, but also in healthcare workers who are indirectly exposed to them. Please, improve the background and insert a paragraph regarding the use of capillaroscopy. I suggest to add alo important references, such as:

1- Medicina 202157, 310.  The Assessment of Endothelial Dysfunction among OSA Patients after CPAP Treatment. https://doi.org/10.3390/medicina57040310

2- Front Pharmacol. 2019 Apr 16;10:360. Innovations in the Assessment of Primary and Secondary Raynaud's Phenomenon.  doi: 10.3389/fphar.2019.00360. 

3- Diagnostics (Basel). 2023 May 29;13(11):1905.  Microvascular Alteration in COVID-19 Documented by Nailfold Capillaroscopy. doi: 10.3390/diagnostics13111905.

4) Introduction. L47-49. The purpose of this study is to add a piece in the definition of a capillaroscopic pattern of ionizing radi- ation damage, by identifying which capillascopic parameters are most affected in  healthcare professionals chronically exposed to ionizing radiation. Please improve the description of study aim, expecially regarding the novelty af these observations.

5) L72-73. Anamnestic and personal data, in- cluding age and gender, of all subjects were collected before capillaroscopic examination. Please specify the presence/absence of Raynaud's phenomenon.

6) 2.3. Capillaroscopic parameters L81-83. Given the absence of a specific capillaroscopic pattern caused by ionizing radiation, ten  morphological qualitative/quantitative parameters were taken into consideration, as-  signing each of them a score on a scale from 0 to 3 (0 = no changes, 1 = <33% abnormal ... Could you please underline the parameters evaluated ( expecially the number of capillaries)? Please support these evaluation with several references, I suggest:

a- J Rheumatol. 2016 Mar;43(3):599-606.  Quantitative Alterations of Capillary Diameter Have a Predictive Value for Development of the Capillaroscopic Systemic Sclerosis Pattern. doi: 10.3899/jrheum.150900.

b- Nailfold capillary patterns in healthy subjects: a real issue in capillaroscopy. Microvasc Res. 2013 Nov;90:90-5. doi: 10.1016/j.mvr.2013.07.001.

7) Figure 1 compares the capillaroscopic findings found in radiation-exposed subjects and 130 healthy controls. Please ameliorate the quality of images c, d, e, f

8) 4. Discussion L140-145.  Healthcare workers employed in the field of radiology and radiotherapy are daily sub- jected to low doses of ionizing radiation, whose impact on long-term health is being  studied. In the context of stochastic damages from radiation and, in particular, for doses  lower than 100 mGy, it is not possible to exclude the onset of cancer, and furthermore,  below this threshold, the incidence of hereditary diseases or tumors seems proportional  to the equivalent dose increment [4]. Please improve the summary of most important results.

9) 5. Conclusions L315-322. We conducted a case-control study aimed at evaluating the biological effects of chronic  radiation exposure in healthcare professionals by capillaroscopic examination of proxi-  mal nail folds. We found that, compared with controls, healthcare workers chronically  exposed to ionizing radiation presented decreased capillary length, increased visibility of  the subpapillary venous plexus, and higher frequency of ectasias, tortuosity, and neoan- giogenesis signs. These capillaroscopic alterations may be clues of radiation damage and  could be part of a future definition of a specific capillaroscopic picture in the early stages  of biologic damage in this group of healthcare workers. Underline the novelty of the study and the possible clinical implications

Author Response

Dear Reviewer,

Thanks for your useful comments and suggestions. Please find below a point-to-point short response to your questions.

1) Abstract. L20- 22. Ten morphological qualitative/quantitative parameters were taken into considera-tion, assigning each of them a score on a scale from 0 to 3 (0 = no changes, 1 = <33% abnormal ca-pillaries, 2 = 33–66% of abnormal capillaries, 3 = >66% of abnormal capillaries, for single magnifi- cation field at 200x). Could you please insert the ten parameters evaluated?Do you evaluate the number of capillaries?

1) We added the ten parameters to the abstract. Yes, we evaluate the number of capillaries.

2) Abstract. L28-30. Conclusions: We found that some capillaroscopic parameters, such as variability in length of capillaries, visibility of subpapillary venous plexus visibility, presence of ectasias, tortuosity and neoangiogenesis signs, are particularly associated with exposure to ionizing radiation in healthcare professionals. Alterations of these parameters may represent capillaroscopic clues of biological damage by chronic radiation exposure in healthcare professionals. Could you please improve the description of possible implications of these observations?

2) We added an explanation on clinical implications of our observations.

3) 1. Introduction L35-37. Exposure to ionizing radiation implies a certain risk of biological damage in the short and long term. Biological effects of ionizing radiations are relevant not only in directly exposed patients, but also in healthcare workers who are indirectly exposed to them. Please, improve the background and insert a paragraph regarding the use of capillaroscopy. I suggest to add alo important references, such as:

1- Medicina 2021, 57, 310.  The Assessment of Endothelial Dysfunction among OSA Patients after CPAP Treatment. https://doi.org/10.3390/medicina57040310

2- Front Pharmacol. 2019 Apr 16;10:360. Innovations in the Assessment of Primary and Secondary Raynaud's Phenomenon.  doi: 10.3389/fphar.2019.00360. 

3- Diagnostics (Basel). 2023 May 29;13(11):1905.  Microvascular Alteration in COVID-19 Documented by Nailfold Capillaroscopy. doi: 10.3390/diagnostics13111905.

3) We added a paragraph on capillaroscopy and the references suggested.

4) Introduction. L47-49. The purpose of this study is to add a piece in the definition of a capillaroscopic pattern of ionizing radi- ation damage, by identifying which capillascopic parameters are most affected in healthcare professionals chronically exposed to ionizing radiation. Please improve the description of study aim, expecially regarding the novelty af these observations.

4) We added an explanation of the novelties of our study compared to previous ones.

5) L72-73. Anamnestic and personal data, in- cluding age and gender, of all subjects were collected before capillaroscopic examination. Please specify the presence/absence of Raynaud's phenomenon.

5) Raynaud’s phenomenon was absent in all subjects. We specified it in the text.

6) 2.3. Capillaroscopic parameters L81-83. Given the absence of a specific capillaroscopic pattern caused by ionizing radiation, ten  morphological qualitative/quantitative parameters were taken into consideration, as-  signing each of them a score on a scale from 0 to 3 (0 = no changes, 1 = <33% abnormal ... Could you please underline the parameters evaluated ( expecially the number of capillaries)? Please support these evaluation with several references, I suggest:

a- J Rheumatol. 2016 Mar;43(3):599-606.  Quantitative Alterations of Capillary Diameter Have a Predictive Value for Development of the Capillaroscopic Systemic Sclerosis Pattern. doi: 10.3899/jrheum.150900.

b- Nailfold capillary patterns in healthy subjects: a real issue in capillaroscopy. Microvasc Res. 2013 Nov;90:90-5. doi: 10.1016/j.mvr.2013.07.001.

6) We added a wider explanation with other references, including these suggested.

7) Figure 1 compares the capillaroscopic findings found in radiation-exposed subjects and 130 healthy controls. Please ameliorate the quality of images c, d, e, f

7) Ok, we did it.

8) 4. Discussion L140-145.  Healthcare workers employed in the field of radiology and radiotherapy are daily sub- jected to low doses of ionizing radiation, whose impact on long-term health is being  studied. In the context of stochastic damages from radiation and, in particular, for doses  lower than 100 mGy, it is not possible to exclude the onset of cancer, and furthermore,  below this threshold, the incidence of hereditary diseases or tumors seems proportional  to the equivalent dose increment [4]. Please improve the summary of most important results.

8) We added an explanation with a new reference.

9) 5. Conclusions L315-322. We conducted a case-control study aimed at evaluating the biological effects of chronic  radiation exposure in healthcare professionals by capillaroscopic examination of proxi-  mal nail folds. We found that, compared with controls, healthcare workers chronically  exposed to ionizing radiation presented decreased capillary length, increased visibility of  the subpapillary venous plexus, and higher frequency of ectasias, tortuosity, and neoan- giogenesis signs. These capillaroscopic alterations may be clues of radiation damage and  could be part of a future definition of a specific capillaroscopic picture in the early stages  of biologic damage in this group of healthcare workers. Underline the novelty of the study and the possible clinical implications

9) We added it to the conclusions.

Round 2

Reviewer 2 Report

The authors have significantly improved the article in content and presentation, which is now suited to the high quality of this journal's readership.